# Epstein-Barr Virus and Multiple Sclerosis: A Convoluted Interaction and the Opportunity to Unravel Predictive Biomarkers

**DOI:** 10.3390/ijms24087407

**Published:** 2023-04-17

**Authors:** Oscar-Danilo Ortega-Hernandez, Eva M. Martínez-Cáceres, Silvia Presas-Rodríguez, Cristina Ramo-Tello

**Affiliations:** 1Multiple Sclerosis Unit, Department of Neurosciences, Hospital Universitari Germans Trias i Pujol-IGTP, 08916 Badalona, Spain; 2Department of Immunology, Hospital Universitari Germans Trias i Pujol-IGTP, Universitat Autònoma de Barcelona, 08916 Badalona, Spain

**Keywords:** acute infectious mononucleosis, B cells, chronic EBV infection, Epstein Barr-Virus, EBV latency programs, genomic studies, immune response, molecular mimicry, multiple sclerosis, predictive biomarkers, serum neurofilament light chain protein

## Abstract

Since the early 1980s, Epstein-Barr virus (EBV) infection has been described as one of the main risk factors for developing multiple sclerosis (MS), and recently, new epidemiological evidence has reinforced this premise. EBV seroconversion precedes almost 99% of the new cases of MS and likely predates the first clinical symptoms. The molecular mechanisms of this association are complex and may involve different immunological routes, perhaps all running in parallel (i.e., molecular mimicry, the bystander damage theory, abnormal cytokine networks, and coinfection of EBV with retroviruses, among others). However, despite the large amount of evidence available on these topics, the ultimate role of EBV in the pathogenesis of MS is not fully understood. For instance, it is unclear why after EBV infection some individuals develop MS while others evolve to lymphoproliferative disorders or systemic autoimmune diseases. In this regard, recent studies suggest that the virus may exert epigenetic control over MS susceptibility genes by means of specific virulence factors. Such genetic manipulation has been described in virally-infected memory B cells from patients with MS and are thought to be the main source of autoreactive immune responses. Yet, the role of EBV infection in the natural history of MS and in the initiation of neurodegeneration is even less clear. In this narrative review, we will discuss the available evidence on these topics and the possibility of harnessing such immunological alterations to uncover predictive biomarkers for the onset of MS and perhaps facilitate prognostication of the clinical course.

## 1. Introduction

Multiple sclerosis (MS) is a complex disease in which different risk factors interact with a background of genetic susceptibility. Currently, Epstein-Barr virus (EBV) infection is considered the main risk factor associated with the onset of MS [1,2]. A number of epidemiological studies have demonstrated that EBV infection confers a high risk for MS, particularly in susceptible individuals [3,4,5,6]. The extent of that risk is nearly 32-fold after primary infection [7]. Nevertheless, it is currently unknown why only a few individuals infected with EBV develop MS despite the worldwide prevalence of asymptomatic infection (nearly 90% of the population) [8]. In this regard, some studies have shown that age at primary infection can modify the risk for MS [2]. Likewise, high levels of anti-EBNA antibodies against the nuclear antigens of EBV also confer a high risk for MS in the long term [6]. A history of acute infectious mononucleosis (AIM)—a major presentation of symptomatic EBV infection—also increases the likelihood of developing MS [9]. Additional studies suggest that when the virus interacts with other risk factors the risk for MS increases further (Figure 1) [10,11,12].

In the largest epidemiological study performed so far, almost 99% of individuals who later developed MS were seropositive for EBV. The infection predated MS onset in susceptible individuals within a time-frame ranging from 0 to 10 years [7]. These findings confirmed previous results from studies carried out in patients at early stages of MS, supporting the role of EBV in disease onset [13,14,15]. Although the exact biological pathways triggering this process are not fully understood, various theories and experimental data suggest that a number of pathogenic mechanisms are likely involved [16,17,18]. At the same time, there is disagreement on several issues related to EBV infection, including the detection of viral particles in the central nervous system (CNS) in post-mortem brain tissue samples of patients with MS [19,20,21,22,23,24], and the role of EVB-reactivation in clinical relapses; with conflicting data across studies [25,26,27,28]. For instance, although some researchers have found in situ activation of latent EBV infection in tissue samples from CNS and meningeal layers of MS patients, others have not. Yet other studies have sought to identify genomic transcripts of EBV-DNA from B lymphocytes circulating in the peripheral blood of patients with active MS [29,30]. All these previous findings highlight the complex and enigmatic role of EBV infection in the onset of MS and perhaps also in the activity of the disease [31].

Other researchers have explored the relationship between certain herpes viruses such as the human herpesvirus type-6 (HHV-6) and EBV in the pathogenesis of MS and the possibility of targeting them by using different antiviral treatments [32]. Similarly, several studies have suggested that EBV may interact with traces of human endogenous retroviruses to trigger the disease onset. However, treating patients with MS for a presumed synergistic infection has not yielded significant results [32,33]. Interestingly, Pender et al. attempted to use EVB-specific T cell therapy to lessen the clinical and radiological activity of MS patients, achieving encouraging results [34]. In this regard, clinical trials are currently underway that seek to evaluate the therapeutic role of autologous EBV-specific cytotoxic T lymphocytes in patients with clinically isolated syndrome (CIS), as well as other antiviral drugs targeting EBV in MS patients [35,36].

## 2. Overview of the Pathophysiology of EBV Infection

EBV belongs to the herpes virus family and is also known as human herpes virus 4 (HHV-4). It has been associated with lifelong asymptomatic infections but also different types of lymphomas and epithelial cancers [8]. The main route of transmission is saliva, and acute infection comes about through the invasion of oropharyngeal epithelial cells as well as naïve B lymphocytes located in the tonsils and around the lymphoid tissue in the Waldeyer’s ring [37]. The virus infects cells by different mechanisms of fusion between viral proteins (gp350/gp42) and specific host cell membrane receptors (CD21/HLA-II) depending on the target cell [38]. Over the decades, attempts have been made to elucidate the association between EBV infection and MS. Various factors have been proposed to play a role, such as the interaction between EBV and endogenous retroviruses, damage in the CNS (the ‘bystander hypothesis’), molecular mimicry and cross-reactivity against myelin peptides, as well as an aberrant cytokine response elicited by EBV infection (for a review, see [17]). Interestingly, molecular mimicry in MS has attracted renewed attention because of recent evidence of cross-reactivity between Anoctamin 2 (chloride-channel protein) and GlialCAM (glial cell adhesion molecule) with certain EBV proteins after primary infection [39,40].

### The Infectious Cycle of EBV

The infectious cycle of EBV is divided into two different stages, the lytic and latent phases [37,41]. During the acute lytic phase, viral DNA acquires a circular shape inside the nucleus of the infected cell, which is known as the episome [42,43]. From this intracellular structure, the virus can influence the regulation of host cell gene expression [37]. The lytic phase is also characterized by the active production of new virions, which bud through the host cell’s membrane instead of bursting the infected cells [44]. The expression of early genes in this phase is also responsible for controlling the host cell’s metabolism and inhibiting the antiviral immune response [45]. Moreover, the lytic phase is likewise divided into early and late gene expression subphases. Structural genes encoding for the viral capsid are expressed only in the late lytic phase [41]. By contrast, during the latent phase, EBV downregulates most of the genes needed for structural proteins, arresting the production of new virions [41]. This phase involves four different types of gene program (0–III), which are characterized by a limited expression of specific viral proteins and microRNAs (Figure 2) [46].

The restricted expression profile of viral genes during each latency program has been mostly associated with the onset of specific cancers. The deregulation of these genes can lead to the emergence of lymphomas from each of the three latency programs of EBV infection. Therefore, it would seem that the microenvironment, the location, and the stage-specific viral transcriptional program define the subtype of lymphoma [47]. Similarly, EBV-infected B cells set on the type III latency program have been related with MS onset in susceptible individuals (Table 1) [48]. Although most of the evidence suggests that chronic infection can be the driving force in MS pathophysiology, the role of the EBV-genes encoded during the acute lytic phase or in other latency programs (i.e., I or II) in the induction MS onset is uncertain. Notably, the infected B cells may recirculate between peripheral and oral compartments, where they can be reactivated, thus triggering viral shedding in MS patients. This suggests that the lytic phase potentially plays a key role in the pathogenesis of MS [49,50].

## 3. EBV and the Risk for MS: Evidence from Genomic Studies

In a comprehensive study, Patsopoulos et al. found that 200 common and uncommon genetic susceptibility variants were likely associated with a hereditability pattern in MS. This study implicated genes related to peripheral immune cells and microglia in the pathogenesis of MS [53]. Based on these findings, the influence of EBV on the risk for MS has recently been explored in genomic association studies [48,54,55,56]. For instance, Afrasiabi et al. compared the transcriptomes from activated B cells and EBV-infected B cells during the latency III program in vitro and in vivo to identify the number of MS risk genes that changed their expression upon infection. In silico data analysis revealed that at least 37 common genetic variants known as single-nucleotide polymorphisms (SNPs) tended to be more highly deregulated in EBV-infected B cells than in the activated B cells, suggesting an important role for infection in the onset of MS [48].

### 3.1. Regulation of MS Susceptibility Genes by EBV-Transcriptional Factors

Moreover, Patsopoulos et al. found that MS risk genes exhibited a certain number of binding sites for transcriptional factors that could be up-regulated by EBV non-structural proteins. For instance, the expression of CD40 and TRAF3 MS susceptibility genes was found to be tightly controlled by EBNA2 [48]. In the virus, the EBNA2 transcripts are encoded by a single exon starting from one of two possible promoters, *Wp* or *Cp*, located within the EBV genome (Figure 2). However, this viral protein cannot bind to human DNA efficiently. Instead, EBNA2 engages specific transcriptional factors inside the host’s cell that are able to recognize specific human DNA sequences [56,57] In this way, EBNA2 can up- or down-regulate the gene expression of MS susceptibility loci depending on the host’s genotype. In another study, EBNA2 was able to bind specific loci in at least five different MS risk genes, of which TRAF3/RCOR1 (rs1258869) and CD40 (rs1883832) alleles were found to be MS susceptibility genes [56]. The TRAF3/RCOR1 (rs1258869) MS-risk allele has been described in other genomic studies [53].

### 3.2. Genomic Control of EBNA2 in Other Chronic Inflammatory Diseases

EBNA2 can also indirectly recognize different alleles (SNPs) across the entire human genome of patients with MS and similar chronic inflammatory diseases such as systemic lupus erythematosus, rheumatoid arthritis, inflammatory bowel disease, type 1 diabetes, and celiac disease [55,58]. This association may also explain the high prevalence of EBV infection in individuals with these autoimmune diseases, highlighting the role of the infection in their pathogenesis [59,60]. Although the molecular weight of EBNA2 is approximately 84 kDA, there is no current evidence that this protein can be secreted as a soluble factor from infected B-cells, nor that it can reach the nucleus of non-infected B cells to control their transcription machinery as a transacting factor.

### 3.3. EBV-microRNAs in the Pathogenesis of MS and Cancer

However, EBV also encodes more than 40 microRNAs, which are small non-coding genetic molecules that can regulate protein expression at post-transcriptional level. Although some EBV microRNAs have been related to MS pathogenesis [54,61], the role of the majority in the disease’s onset is still unknown [62]. Various EBV microRNAs have been implicated in the commencement of carcinomas after years of chronic infection. Therefore, it may be that EBV exerts a broad “genetic manipulation” effect over the host’s gene expression. This relationship between EBV and human DNA may be conceived as the consequence of historical migrations in ancient times and other evolutionary pressures acting in parallel as has been described regarding the role of other microorganisms in the history of human evolution [63]. Although all previous insights arise from studies performed on B cells infected by EBV—which are the main reservoir for the virus [64]—additional mechanisms have been implicated in the interaction between EBV and the immune system. For instance, a defective control of EBV replication by cytotoxic T lymphocytes (CTLs) of the host has been reported in individuals at early stages of MS [65,66]. On the other hand, other studies have shown that CTLs can mount a robust response against infected B cells during acute infection [67].

EBV-DNA lacks its own methylation system, histones, or other epigenetic regulators. However, the virus lifecycle is synchronized by specific epigenetic mechanisms that control gene expression during the lytic and latency programs in both EBV-infected B cells and epithelial cells [37,46]. Such mechanisms are driven by viral promoters and non-coding microRNAs which have been implicated in the onset of lymphomas and gastric carcinomas. Significantly, such regulators have been recently linked to the pathogenesis of MS. In one study, seven MS susceptibility genes were found to be up-regulated by at least fifteen EBV-derived microRNAs [68]. All previous findings at the genomic level point to another critical link between EBV infection and the pathogenesis of MS aside from the classic hypothesis [17].

## 4. EBV Decides the Fate of Infected B Cells and Likely of Autoreactive B Cells

During primary infection, almost 50% of the entire pool of peripheral memory B cells can be infected by EBV, with a subsequent rapid decline in the number of infected cells a few weeks later [69]. Moreover, recent evidence shows that immediately after acute infection (i.e., in the lytic phase) EBV begins to reprogram the infected B cells, triggering important changes in the expression of specific host signaling proteins on the cell membrane [70]. Subsequently, the infected B cells express the virus’s genetic latency program III, which triggers active proliferation of plasmablasts and the assembly of new virions [51]. Nevertheless, the precise molecular steps involved in the development of MS in some individuals and chronic lymphoproliferative disorders or systemic autoimmune diseases in others are unknown. These variations may be related to individual genetic susceptibility or else certain EBV virulence factors. For instance, specific genes encoding oncogenic proteins during EBV chronic infection have been extensively associated with the onset of lymphomas in susceptible patients (Table 1) [42,71]. However, patients with MS do not bear an increased risk for lymphoproliferative disorders that can be directly linked to EBV infection.

### 4.1. The Role of Other EBV-Encoding Products in the Pathogenesis of MS

As mentioned above, recent evidence suggests that EBNA2 is likely implicated in modulating the transcriptional rate of susceptibility genes for MS [56]. Nevertheless, the role of other proteins expressed by EBV in the pathophysiology of MS such as LMP1 and -2 [71] and those from the lytic phase is currently unknown [52]. In addition, certain microRNAs encoded by EBV may also modify the immune response, easing the onset of cross-reactivity against myelin peptides [72].

### 4.2. EBV and Autoreactive B Cells in MS

There is no consensus on whether or not EBV infects autoreactive B cells in patients with MS [73]. The genomic transcripts of EBV cannot be consistently isolated from polyclonal activated B cells circulating in the peripheral blood of patients with active MS not even using high-resolution techniques [30]. Although important differences in detection techniques may explain some of these contradictory results [74], there may be room for another explanation. For instance, it has been suggested that autoreactive B cells need first to receive an activation signal sent by follicular dendritic cells (FDCs) via a HLA class II antigen presentation pathway from EBV-epitopes attached to their membranes. In fact, this is the normal route for the activation of B cells in the germinal center (GC), after they receive the T cell stimuli and before they reappear in peripheral tissues [75,76]. Likewise, this possibility also supports the role of molecular mimicry in the pathogenesis of MS [39,40,77].

Various studies have shown that EBV guides the infected B cells directly into the GC inside secondary lymphoid organs. There, under the latency III program, the infected cells may serve as the major source of lytic- or latent-encoded viral antigens to the FDCs. Inside the GC, these cells may activate autoreactive B cells before migrating to the CNS to amplify the immune response [37,78]. In this connection, one study found that EBV can infect and transform FDCs in vitro [79]. Likewise, it has been shown that deregulated FDCs can rescue autoreactive B cells from apoptosis, favoring the onset of autoimmunity [78]. In the GC, EBV-infected B cells appear to undergo the normal process of high-affinity maturation and somatic hypermutation (SHM) as non-infected B cells do when they are primed with specific antigens in peripheral tissues [80]. However, B cells infected by EBV apparently do not complete the SHM process. Instead, they may leave the GC as long-lived plasma cells or circulate in peripheral blood as quiescent memory B cells under the control of the EBV latency programs (0/I). Interestingly, from peripheral blood they may also re-enter the GC [81]. It is important to point out that continuous circulation of pathogenic non-infected (autoreactive) B cells between peripheral and CNS compartments has been shown to occur in MS [30,82]. This also suggests that constant activation of autoreactive B cells may take place in both the deep cervical lymph nodes (DCLNs) and inside the CNS, which may be critical for prolonging the inflammation observed in MS (Figure 3) [83].

## 5. Where Does EBV Actually Reside in Patients with MS?

Neuropathological studies performed in patients with MS have shown relatively little inflammation in focal demyelinating lesions, especially within the newly formed plaques at early stages. Instead, extensive areas of apoptotic oligodendrocytes and numerous CD68 positive microglial cells are often seen [84,85]. Such limited levels of inflammation inside the new plaques may be explained to some extent by the lack of initial activation of autoreactive B cells in the DCLNs and non-cervical lymph nodes during early stages of inflammation in the CNS [75,76]. Although the presence of EBV in biopsies of the white matter of patients with MS is still controversial, the virus may orchestrate an abnormal immune response by mechanisms other than primary CNS infection. For instance, if a pool of long-lived B cells infected with EBV actually resides within the DCLNs [86,87] and/or inside the tertiary lymphoid organs rather than in the CNS itself [88,89], such tissues may still provide a link between EBV infection and the nurture of self-reactivity in MS-susceptible individuals.

However, this supposition is still difficult to prove due to significant variations in the sensitivity and specificity of the histopathology techniques that were used in previous studies [90]. Recently, new immunomodulatory cell therapies aimed at down-regulating self-reactivity in patients with MS have been applied directly in the cervical lymph nodes [91]. This new therapeutic approach is aligned with evidence showing that continuous drainage of cerebrospinal fluid, CNS-antigens, and resident antigen-presenting cells (APCs) may take place from the CNS directly into DCLNs [92,93,94]. This interchange of antigens and debris seems to occur throughout highly specialized anatomical structures located at the skull base of humans (Figure 3) [93].

## 6. EBV Infection and the Chance of Discovering Predictive Biomarkers for MS

The recognition of EBV infection as the sine qua non condition preceding the onset of MS opens new avenues for studying the pathophysiological mechanisms occurring in the “incubation period” of this inflammatory disease, known as prodromal MS. The goal of research into prodromal MS is to discover predictive biomarkers for onset of the disease in susceptible individuals [95,96]. For instance, potential predictive biomarkers related to EBV infection could be measured in patients with acute infectious mononucleosis (AIM), especially in those with other concomitant risk factors for MS, like smoking, excess weight, low vitamin D levels, genetic predisposition, or young age [6,17]. Nevertheless, in practical terms this research goal is currently difficult to pursue given, first, the low incidence of MS cases and, second, the disease’s complex genetic hereditability pattern, which only explains 24% of new MS cases [56]. Consequently, even if predictive biomarkers for MS onset were identified, it would be challenging to select the most appropriate candidates for screening [97].

### 6.1. EBV and Neurodegeneration in MS

On the other hand, longstanding debate exists around neurodegeneration in MS. Some researchers believe that already at baseline MS is a neurodegenerative disease that is rendered covert by the extensive inflammatory component typically present at early stages of the disease [98]. Contrastingly, others consider neurodegeneration to be the final consequence of chronic inflammation [99]. Since little is known about the pathogenic events that happen in prodromal MS—before the first symptoms appear—advocating one theory over the other is difficult because of the amount of evidence supporting both views [98,99]. Currently, the role of EBV infection in brain atrophy and disability progression is unknown. However, this possibility cannot be completely dismissed [100,101]. Indeed, some researchers have found that EBV chronic infection may trigger mitochondrial dysfunction in infected patients without MS [102].

In this regard, two different studies have shown that high levels of serum neurofilament light chain protein (sNFL) can be detected in serum samples of asymptomatic individuals, before the onset of MS [7,103]. Intriguingly, in the study performed by Bjornevik et al. sNFL levels started to increase only after the EBV seroconversion of individuals who later developed MS [7]. It is well known that increased titers of sNFL reflect axonal damage in patients with MS at baseline and likely during relapses [104]. Further research has also shown that sNFL levels may be used as a potential biomarker of subclinical disease activity and cognitive decline in MS patients [105,106]. These findings help to illustrate how the interplay between EBV infection—the most important risk factor for MS—and the immune system can be harnessed to study new potential biomarkers at different stages of the disease.

### 6.2. EBV Vaccination and Anti-CD20 Therapies

It would be of considerable interest to evaluate the impact of mass vaccination against EBV on the incidence and clinical course of MS. In addition, understanding the influence of vaccination on the frequency of lymphoproliferative disorders and systemic autoimmune diseases would probably help us to clarify the role of EBV infection in these inflammatory diseases. Indeed, in order to explore this promising approach, clinical trials are currently testing different EBV-vaccine candidates [107,108]. Other researchers have proposed that the efficacy of anti-CD20 therapies in patients with MS may be also related with removing EBV-infected B cells from peripheral blood [109]. In this regard, it would be of interest to evaluate the percentage of infected B cells that can be directly targeted by anti-CD20s. However, it is also important to point out that anti-CD20 therapies have shown only moderate efficacy in the treatment of patients with primary progressive MS [110].

## 7. Conclusions

Strong epidemiological evidence supports the association between EVB infection and the onset of MS. However, the molecular mechanisms involved in this process and in the maintenance of inflammation, as well as EVB’s contribution to neurodegeneration—if any—have not been fully elucidated and warrant further investigation. That said, the available data suggest that EBV may dysregulate the immune system of individuals at high risk for MS by means of a wide range of disease-driving mechanisms. These pathogenic pathways encompass the epigenetic control of MS susceptibility genes by EBV virulence factors as well as other intricate molecular routes such as the putative polyclonal activation of autoreactive B cells inside DCLNs by cross-reactivity with EBV antigens (molecular mimicry), among others. Although EBV has also been linked to the pathophysiology of lymphoproliferative disorders, epithelial cancers, and systemic autoimmune diseases, at present, there are gaps in our understanding of this association. Therefore, the specific molecular entryways involved in the onset of MS in some individuals and lymphomas in others are also unknown. Hopefully, in the near future it will be possible to use various epigenetic and immunological disturbances triggered by the virus after interacting with the host cells as predictive and reliable biomarkers not only for MS onset but also to anticipate the course of the disease. In the meantime, several clinical trials are underway to explore the benefit of targeting EBV infection in MS patients.

## Figures and Tables

**Figure 1 ijms-24-07407-f001:**
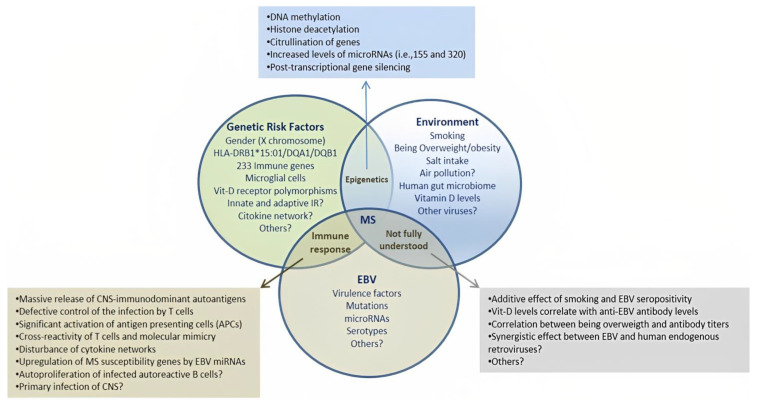
Interaction among different risk factors for multiple sclerosis (MS). A number of genetic, epigenetic and environmental factors have been associated with MS. However, EBV infection is now considered an essential condition for disease onset. Recent evidence also suggests that EBV may duplicate the risk of MS when other risk factors coexist in the same individual. Although there is a lack of agreement regarding some of the factors listed in the diagram, these interactions can be further studied through various methods using biological specimens taken from susceptible individuals. This may also provide a unique opportunity to disentangle the origins of this complex and enigmatic disease. IR: immune response.

**Figure 2 ijms-24-07407-f002:**
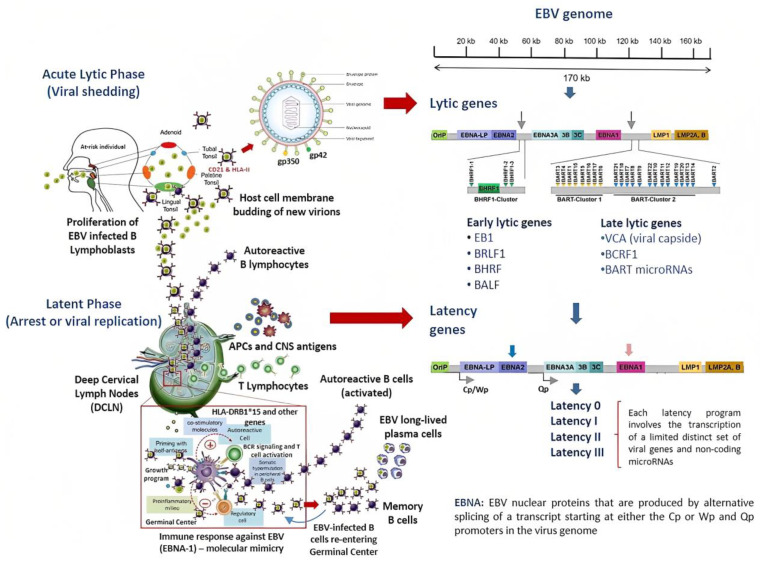
Life cycle and EBV-gene expression during acute and chronic infection. EBV is transmitted by saliva. Epithelial cells and naïve B cells in the local lymphoid tissue can be infected through specific receptors. Early after infection, EBV starts reprogramming the B cell, inducing its rapid proliferation and the assembling of new virions which bud from the cell membrane. These events can occur during acute infection (lytic phase) or after EBV reactivation. This phase is also characterized by the expression of a specific set of EBV genes. After this stage, the virus sequesters the infected B cells and directs them towards the germinal centers inside the secondary lymphoid organs. There, during the latent phase of infection, EBV decides the fate of infected cells independently from the BCR signaling pathway or the activation by T cells. The latent phase is made up of four latency viral programs each defined by the expression of a unique set of EBV genes. However, the entire EBV genome is transcribed only in the type III latency program, which is thought to be related with MS onset. Finally, it may be postulated that EBV-infected B cells can also re-enter the germinal center. There, EBV may execute fine-tuning in the host gene expression of infected B cells while it efficiently manipulates the antiviral immune response. These mechanisms have been closely linked with the onset of lymphoproliferative disorders. BCR: B-cell receptor.

**Figure 3 ijms-24-07407-f003:**
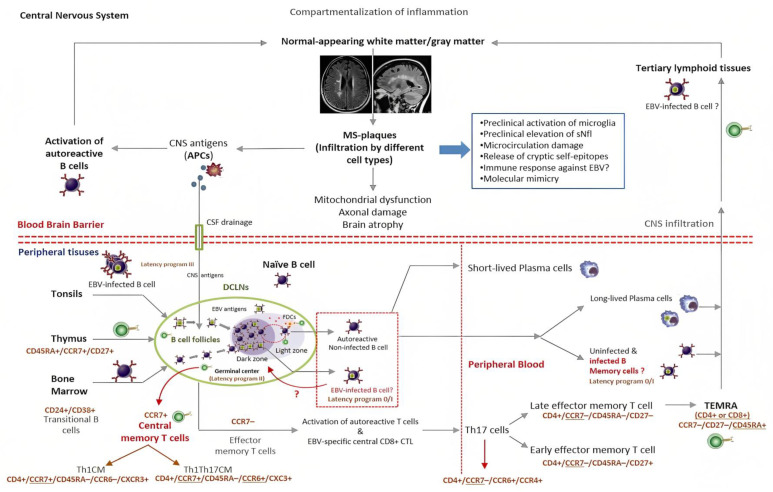
Theoretical model for the role of EBV in the pathogenesis of MS (simplified view). EBV may act as the main trigger of the inflammatory cascade that starts demyelination in the CNS of susceptible individuals. In the upper half, the activation of autoreactive B cells, the continuous drainage of CNS-antigens, the recruitment of inflammatory cells, and the potential role of tertiary lymphoid tissues in the pathophysiology of MS are shown. In the bottom half, the DCLNs are considered to be the sites where autoimmune responses begin. These organs may serve as the “meeting point” for EBV-infected B cells, autoreactive B cells, T cells, and self-antigens drained from CNS through specialized structures located at the skull base. In the germinal center, the virus defines the fate of infected B cells as either circulating quiescent memory B cells or long-lived plasma cells. On the other hand, while transiting from the dark to the light zone of the germinal center and after receiving a second activation signal by FDCs and T cells, the autoreactive B cells exit the lymphoid organs to carry out diverse functions in the CNS. This representation is also consistent with the hypothesis that the molecular mimicry between viral epitopes and myelin self-antigens is one of the main disease mechanisms involved in the pathogenesis of MS. Similarly, the diagram suggests that continuous stimulation of the immune system with self-antigens and viral products may explain the chronic disease activity in MS if EBV latent infection is assumed (clockwise). Cell membrane biomarkers of different lymphocyte subpopulations that may be involved in the interaction between EBV and the immune system can be assessed by flow cytometry. APC: professional antigen-presenting cells. CM: central memory cells. CTL: EBV-specific cytotoxic T cells. DCLNs: deep cervical lymph nodes. FDCs: follicular dendritic cells. GC: germinal center. sNLF: neurofilament light chain. TEMRA: effector memory cells re-expressing CD45RA.

**Table 1 ijms-24-07407-t001:** Specific patterns of EBV gene expression during the latent phase of infection and their association with malignancies and MS.

Latency Program	Alternative Name	Gene Products ^1^	Site In Vivo	Stage of Normal B-Cell Development	Events	Biomarkers	Associated Disease ^1,2^	Ref
**0**	Latency program 0	EBERs BART microRNAs (non-coding genes)	Peripheral circulation	Latently infected memory B cells	Downregulation of most viral protein-encoding genes suppressing the production of virions	None	None	[37]
**I**	Latency program I	**EBNA1** ^3^ EBER1/2 RNA BART microRNAs	Peripheral circulation	The homeostatic proliferation of memory B cells is not driven by the virus	Replication of EBV genome harnessing the mitotic cycle of memory B cells	Anti-EBNA1 IgG Anti-VCA IgG	BL	[42]
**II**	Default program	**EBNA1** ^3^ EBER1/2 RNA BART microRNAs LMP1 LMP2A/B	Tonsil germinal center (GC) and lymph nodes	Naïve B cells infected by EBV gain access into the GC for the normal process of differentiation	EBV set a limited transcriptional program that rescues EBV-DNA into the memory B cell compartment where viral DNA persists as the episome	Anti-EBNA1 IgG Anti-VCA IgG	HL T/NK-cell malignancies Epithelial malignancies NFC	[41,51,52]
**III**	Growth program	**EBNA1** ^3^ **EBNA2** **EBNA3A, 3B, 3C** EBER1/2 RNA EBNALP BART microRNAs **LMP1** **LMP2A/B**	Tonsil GC/Lymph nodes	Activation of naïve B cells infected by EBV In this lytic phase, B cells become proliferating B-blasts before entering the GC	Full expression of EBV proteins CTL trigger a strong immune response to suppress EBV-infected B cells Selective silencing of EBV genes upon unknown individual and environmental conditions	Anti-EA IgM Anti-EBNA1 &-2 IgM Anti-VCA IgM/IgG CTL	DLBCL Post-transplant lymphoproliferative disorders CNS lymphoma MS?	[41,42,47,52]

^1^ See the glossary section for abbreviations. ^2^ EBV-gene expression is tightly regulated in a tissue-specific manner. ^3^ EBNA1 is expressed from promoter *Qp* in the latency programs I/II and from the *Cp* promoter in the growth program III from the EBV’s circular shaped DNA inside the cell nucleus (episome). This protein allows the viral genome to replicate at the same time with the host cell. The main gene transcripts of EBV are shown in bold letters.

## Data Availability

The data that support the findings of this review are available by searching for the following MeSH terms in PubMed: Acute infectious mononucleosis; B cells; chronic EBV infection; Epstein Barr-Virus; EBV latency programs; genomic studies; immune response; molecular mimicry; Multiple Sclerosis; predictive biomarkers; serum neurofilament light chain protein.

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
