# Peer review of "Epstein-Barr Virus and Multiple Sclerosis: A Convoluted Interaction and the Opportunity to Unravel Predictive Biomarkers"

_ijms, 2023, doi:10.3390/ijms24087407_

Round 1
Reviewer 1 Report
I have not any Comments and suggestions for Authors.
Author Response
Dear reviewer, thank you so much.
Reviewer 2 Report
This manuscript is a review of important research in which the authors are compiling research aimed at discussing the topics of Epstein-Barr virus infection as a main risk factor for MS and the molecular mechanisms of such including molecular mimicry, bystander damage, abnormal cytokine networks, and coinfections with other viruses. The authors present interesting review evidence for association for EBV and MS but clarity and flow is an issue throughout the manuscript. There seems to be many statements made but tying them to the next is rough. Further, if over 90% of the world population has had EBV infection, it might be helpful if the authors clarify the significance of EBV seroconversion and MS onset early on in the manuscript.
Point 1: Spell check throughout the manuscript including the figures.
Point 2: In line 110. Active EBV or MS?
Point 3: Section 2 starting in the second paragraph is confusing (line 143). Figure 2 and its legend could use a little more work describe the life cycle as it appears in the figure. In line 169 it says that the entire EBV genome is transcribed only in latency iii. But the latency phase involves four programs each characterized by limited expression of proteins which is confusing. Why is this important. I think bringing Table 1 together with figure 2 will help clarify. Also line 172 seems to be missing a word.
Point 4: Will table 1 be on one page?
Point 5: What happened to the line numbering after the table starting on page 9?
Point 6: On page 9 there is a paragraph discussing EBNA2 indirectly recognizing SNPs concerning other autoimmune diseases which I don’t understand where the statement “may explain the high prevalence of EBV infection in individuals with these autoimmune diseases” strengthens the argument for MS being presented here. Please explain why EBV infection is important for all these diseases.
Author Response
Dear reviewer.
Thank you so much for your comments and suggestions. We really appreciate that. Next, we will response point by point of your queries:
Point 1: Spell has been check throughout the entire manuscript
Point 2: We meant MS (corrected).
Point 3: Spell of figure 2 and legend has been cheked. EBV lifecycle is defined by four different programs (0-III). We highlight this phase in the review article because there are recent studies showing viral shedding in MS patients, which means reactivation (Ref 49, 50). Therefore, it would be interesting to study the role of these programs in the pathogenesis of MS.
Point 4: I may ask to the editorial office to help us to organize Table 1 in just one page.
Point 5: We do not known what could had happened about the numbering at the begining of each line after page number 9. This is something that we believe was assigned automatically after uploading the paper.
Point 6: high prevalence of EBV infection has been also described in patients with systemic autoimmune diseases (AIDs) such as rheumatoid arthritis, systemic lupus and sjogren syndrome. A recent study reinforces the posibility of a similar genetic mechanism shared between MS and AIDs regarding EBV infection (Ref 56, 57).
Reviewer 3 Report
The article by Ortega-Hernandez et al., titled “Epstein-Barr virus and Multiple Sclerosis: A convoluted interaction and the opportunity to unravel predictive biomarkers” contains interesting perspectives is to understand the role of EBV in the pathogenesis of multiple sclerosis (MS). The authors have shown in-depth knowledge of the subject. However, I have a few concerns which are listed below for the authors to improve this manuscript –
Major Concerns-
Since this convoluted interaction doesn’t seem to be conclusive with clear take home message, such as what kind of reliable biomarkers specifically can be used for the predictive onset of MS?
1- In the discussion section, the authors are suggested to mention the highlights of this review or what are the differentiative factors from the reviews which were published in the last 2-3 years?
Correction-
1- In the conclusion section, the authors are suggested to mention the highlights of this review or what is the take home message?
2- Authors are advised to reframe the structure of the manuscript in order to avoid very long paragraphs and encouraged to add sub-headings under long headings such as section 3, 4 and 6.
3- Referencing in the table needed to be improved to avoid confusion. References need to be mentioned at respective places in the table for individual study. Also the numbering system is very confusing between references and footnotes in the provided table.
Minor Concerns-
1- In the line 138; Authors are suggested to use brackets inside parentheses to create a double enclosure in the text.
2- Authors are suggested to keep uniform format in the entire manuscript.
3- Figure 1 fonts are too small to read in the printed manuscript. Authors are suggested to make the text easily visible.
4- Similar problem in figure 2, fonts are too small to read in the printed manuscript. Authors are suggested to change the orientation of the figure to make the text easily visible.
Author Response
Dear reviewer.
Thank you so much for your comments and suggestions. We really appreciate that. Next, we will response your queries point by point:
Point 1: we have improved the wording of the conclusions.
Point 2: We have reframed the structure of the manuscript and added sub-headings under each section.
Point 3: We have modified the table in order to avoid confusion between references and footnotes.
We have modified all aspects of the minor concerns section in your comments.
Reviewer 4 Report
The review manuscript aimed at giving an overview of the relationship between Epstein-Barr virus (EBV) and Multiple Sclerosis (MS). The authors discussed the available evidence on the role of EBV infection in the MS and the possibility of harnessing immunological alterations to uncover the predictive biomarkers for the onset of MS and likely for the prognosis of the clinical course. The authors discussed the pathophysiology of EBV infection, genomic evidence of EBV and risk for MS, the effect on infected B cells and autoreactive B cells by EBV, localization of EBV and potential predictive biomarkers.
The manuscript is clear, and relevant for the field. The manuscript is very informative and the language is clear.
The topic is quite original. However, the authors did not discuss enough why EBV needs special attention in MS. If the authors can discuss more about current status of MS related research and why EBV needs special attention, it would be helpful.
Line 76, “~90 of population”. The author may mean “about 90% of the population”. The figure layout can be further improved for easy understanding.
Author Response
Dear reviewer.
Thank you so much for your comments and suggestions. We really appreciate that. Next, we will response your queries point by point:
We have improved the wording and the main ideas in the paragraph. We have also improved the figure layouts.
Reviewer 5 Report
Very interesting and clear manuscript. A very "hot topic" of interest for MS experts. Very nice figures and clear table.
Some minor comments:
-I would specify the type of review (I suppose a Narrative Review), and the methodology of paper selection
-Finally I would recommend using the correct MDPI template according to the "instruction for Authors" section.
Thank you
Author Response
Dear reviewer.
Thank you so much for your comments and suggestions. We really appreciate that. We have made all changes that you have suggested for the review. Methodology was made by MeSH terms search not throughout a systematic review.
Reviewer 6 Report
This is a clearly written and well-organized review. It summarizes the latest findings on the relationship between EBV and MS. I consider that this review might be relevant for the audience to read and to take into consideration for their future research and I suggest to accept it in the present form.
Author Response
Dear reviewer, thank you so much.
Round 2
Reviewer 3 Report
After revision manuscript is improved. There are no further comments from my side.